# The effects of provisional resin cements on the color and retentive strength of all-ceramic restorations cemented on customized zirconia abutments

Seyede Mina Salehi Dehno[1], Rashin Giti[1]*, Mohammad Hassan Kalantari[1], Farhad Mohammadi[2]

**1** Department of Prosthodontics, School of Dentistry, Shiraz University of Medical Sciences, Shiraz, Fars, Iran, **2** Department of Pharmaceutics, School of Pharmacy, Shahid Sadoughi University of Medical Sciences and Health Care Services, Yazd, Iran

* giti_ra@sums.ac.ir

## Abstract

This study aimed to evaluate the effects of two types of provisional resin cements on the color and retentive strength of two different all-ceramic restorations cemented onto customized zirconia abutments. Forty-two crowns were made of monolithic zirconia and lithium disilicate ceramics ($n = 21$ per group) and cemented on customized zirconia abutments by using two provisional resin cements of TempBond Clear and Implantlink Semi, and TempBond serving as the control (n = 7 per cement subgroup). The specimens' color was measured before and after cementation and after thermocycling. The color difference was calculated by using CIEDE2000 formula ($\Delta E_{00}$). The tensile force was applied to assess the retentive strength. Kruskal-Wallis, Dunn's post-hoc, and Mann-Whitney non-parametric tests were used to compare $\Delta E_{00}(1)$ and $\Delta E_{00}(2)$ and two-way ANOVA followed by one-way ANOVA and Tukey's HSD post hoc test and T-test were used to compare retentive strength between subgroups. In the lithium disilicate group, $\Delta E_{00}$ of the control subgroup (Temp-Bond) was significantly higher than that of Implantlink Semi cements subgroup ($P = 0.001$). But, in the monolithic zirconia group, $\Delta E_{00}$ of the control subgroup (TempBond) was significantly higher than that of Implantlink Semi ($P = 0.020$) and TempBond Clear cements ($P = 0.007$). In the monolithic zirconia group, the control subgroup (TempBond) was significantly more retentive than TempBond Clear ($P = 0.003$) and Implantlink Semi cement ($P = 0.001$). However, in the lithium disilicate group, Implantlink Semi cement was significantly more retentive than TempBond Clear ($P = 0.019$) and TempBond (control) ($P = 0.001$). The final color of both restorations was significantly affected by the provisional resin cement type. The retentive strength was influenced by both the type of cement and ceramic.

## Introduction

An implant is a durable and successful option for esthetic reconstruction and function of the missing teeth [1]. They are attached to the abutments with either screws or cements. The

**Data Availability Statement:** All relevant data are within the manuscript and its Supporting Information files.

**Funding:** The Vice-Chancellry of Shiraz University of Medical Sciences for supporting this research (grant #21860). The funders had no role in study design, data collection and analysis, decision to publish, or preparation of the manuscript.

**Competing interests:** The authors have declared that no competing interests exist.

clinical success of cement-retained restorations mainly depends on the retention [2], which is affected by many factors like the geometry of abutment preparation, taper, abutment height, surface area and roughness, luting agent [3], as well as the abutment and coping material. While zirconia copings are said to have higher mean retention than metal copings [4], evidence has shown that the abutment material does not affect the copings dislodgement resistance, regardless of the type of cement [5].

The cement type considerably affects the retention of restorations. Specially-formulated cements have been introduced for the cementation of implant-supported prostheses. New provisional resin cements are reported to be more retentive than the zinc oxide-eugenol and non-eugenol cements while precluding the drawbacks [6, 7]. Another study documented that non-eugenol temporary resin cement had significantly higher tensile strength than the non-eugenol zinc oxide cement and resin-based acrylic urethane cement [8].

New ceramic materials like yttrium-stabilized zirconium oxide polycrystal are popular for their premium mechanical properties and tooth-like color [9]. The final color of restorations is affected by numerous factors such as the material and color of the substrate, cement, thickness and shade of zirconia coping and veneering ceramic, and laboratory process [10]. Capa et al. asserted polycarboxylate cement to be better than resin cement for cementation of zirconia restorations on titanium abutments [11]. Another study reported that the resin cement caused the most unacceptable changes in color and translucency of monolithic zirconia [12].

Although new provisional resin cements are claimed to have better properties than their conventional counterparts [7], limited information is available about their effect on the final color and retentive strength of implant-supported ceramic restorations. Given the lack of studies on the effects of specific types of provisional resin cements on the retentive strength and final color of aged implant-supported all-ceramic restorations, the present study aimed to assess the effect of TempBond Clear and Implantlink Semi provisional resin cements used to attach the crowns onto customized zirconia abutments on the retentive strength and final color of implant-supported all-ceramic restorations before and after aging.

The null hypothesis was that the type of provisional resin cement and type of ceramic restoration would not affect the final color and retentive strength of implant-supported all-ceramic restorations.

## Materials and methods

### Fabrication of customized zirconia abutments

This experimental in-vitro study involved 6 maxillary right central incisor customized zirconia abutments (ZrGEN; AnyOne; model: AAOIPR4525; Megagen, South Korea). Six implant laboratory analogs (Lab Analog; Blue color; Megagen, South Korea) were embedded in autopolymerizing acrylic resin blocks (Acropars Re; Marlic; Iran) by using a dental surveyor (Marathon-103; Saeyang; Korea).

A single ceramic superstructure was computer-aided designed ([CAD], CAD design software; 3shape, Denmark) and milled out of high translucency zirconia blanks in A2 shade by using a laboratory milling machine (Cori Tec 340i; Imes-Icore GmbH, Germany) with 6-mm axial wall height, 11-degree axial taper, 1-mm deep shoulder finishing line, and 2-plane reduction of labial surface (Fig 1).

Titanium inserts were screwed to the implant analogs and tightened to 35 Ncm torque by a manual torque wrench (Megagen, South Korea). The superstructures were bonded to titanium inserts by using dual-cure self-adhesive resin cement (RelyX$^{TM}$ U200; 3M ESPE, USA) according to the manufacturer's instructions. The six abutments were used for 60 crowns and cleaned between each run. Removal of the temporary cement consisted of gross removal with explorer,

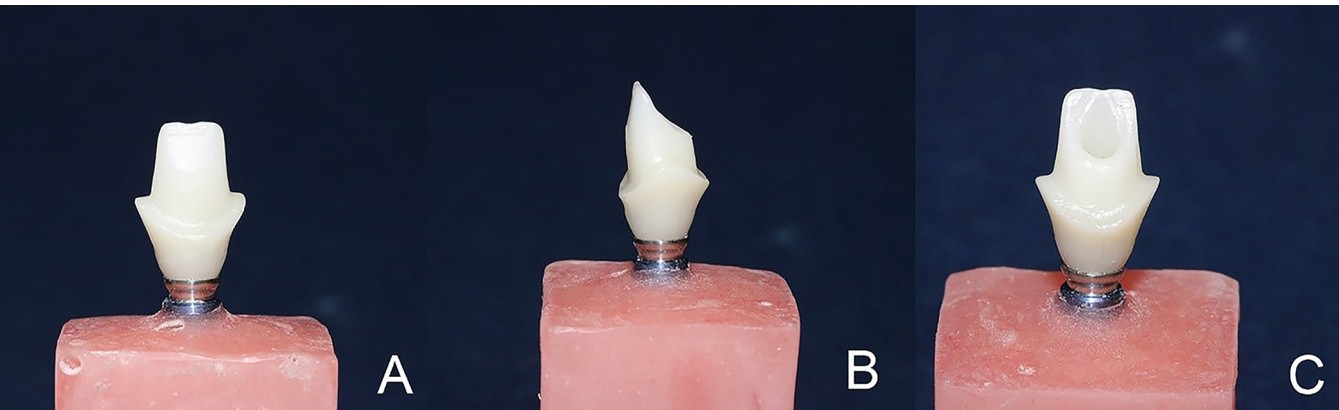

**Fig 1. Milled customized zirconia abutment superstructure.** A: buccal surface, B: proximal surface, C: lingual surface. The bonding surface of the titanium implant insert (ZrGEN; AnyOne; model: AAOIPR4525; Megagen, South Korea) was abraded with 50-μm aluminum oxide particles at a 10-mm distance for 10 seconds (0.4 MPa). The inner surface of zirconia abutment superstructures was also subjected to airborne particle abrasion with 30-μm silica-coated aluminum oxide (Rocatec Soft; 3M ESPE, USA).

ultrasonic bath with 70% ethanol for 15 minutes, and application of 37% phosphoric acid for 30 seconds for complete removal of the cement remnants. The specimens were then rinsed and dried.

## Fabrication of all-ceramic crowns

A full contour right maxillary central incisor crown with a projection on its incisal edge was designed by using CAD software. A 2-mm (diameter) hole was considered in the middle of this projection to attach the crowns to the universal testing machine (Fig 2). The crowns were designed in standard dimensions, 40-μm cement space, and an average 1-mm thickness in the mid-third of the labial surface, where color measurement was done.

The crowns were all sandblasted with 50-μm aluminum oxide particles from a 10-mm distance for 10 seconds, steam-cleaned, and dried with oil-free water. The abutments received no surface treatment and were only steam-cleaned and dried with oil-free air before cementation.

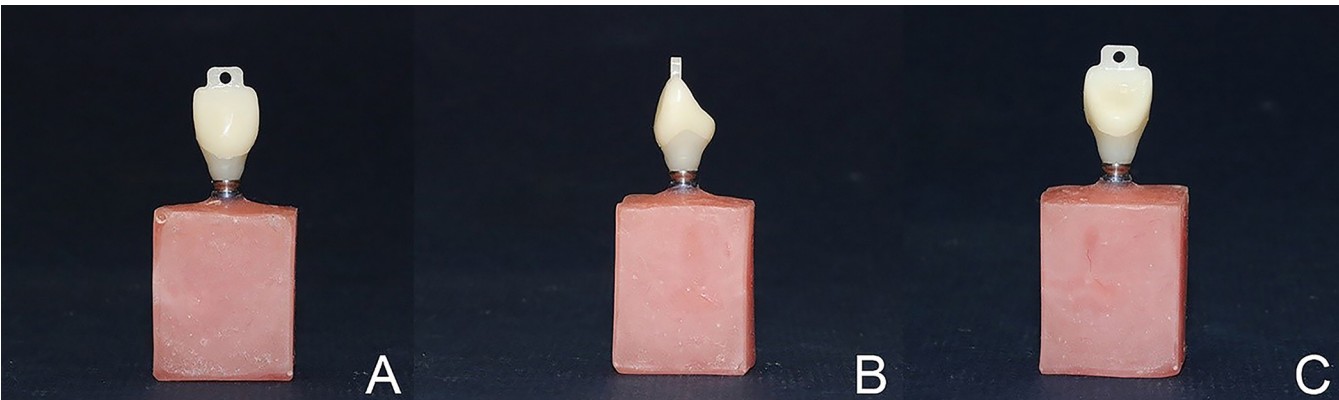

**Fig 2. A full contour right maxillary central incisor crown with a projection on its incisal edge.** A: buccal surface, B: proximal surface, C: lingual surface. Forty-two ceramic crowns (n = 21 per group) were fabricated with either ultrahigh-translucent monolithic zirconia in A2 shade (ZIRAE; SHT Preshaded Zirconia Block, Nanjing Zirae Advanced Material Co, China) by using a laboratory milling machine (Cori Tec 340i; Imes-Icore GmbH, Germany) or high-translucent lithium disilicate ceramic in A2 shade (IPS e.max Press; Ivoclar Vivadent, Germany). For making lithium disilicate crowns, wax patterns were prepared by a 3D printer (Solidscape D76+; Solidscape, USA) based on the previously CAD-designed shape and dimensions. The wax patterns were sprued, invested, burned out; and lithium disilicate was pressed into the burned-out molds.

## Color measurement and cementation

Having seated the crowns on the abutments, the shade was measured at the mid-third of the crowns by using a spectrophotometer (VITA Easyshade; VITA Zahnfabrik, Germany) on a 3-mm area. The device was calibrated according to the manufacturer's instruction before each use. To replicate the conditions of spectrophotometry for all specimens and preclude any external light, a silicone putty material (Speedex; Coltène, Switzerland) was molded to the spectrophotometer [13]. Shade parameters (L, a, b) were measured 3 times and the mean value was recorded. The spectrophotometer output was accorded to a 2-degree standard observer and D65 illuminant.

The crowns made with each ceramic system were randomly divided into 3 subgroups (n = 7) to be cemented with provisional dual-cure resin cements of Implantlink Semi (Semi-permanent Implant Cement; DETAX, Germany) and Temp-Bond Clear (Temporary crown and bridge cement; Kerr corporation, Italy); while TempBond (TempBond; Kerr, Italy) served as the control (Table 1). The screw access hole of the customized zirconia superstructure was filled with cotton and composite resin and cemented according to the manufacturer's instructions. To cement with Temp-Bond Clear and Implantlink Semi, the automix tip was placed onto the syringe, the abutment surface and the internal surface of restoration was completely dried, and a thin layer of the mixed cement was applied to the internal surface of the restoration. The crowns were seated on the abutments with firm finger pressure for 10 seconds and subjected to a controlled axial load of 20-N for 10 minutes. It was different from ADA specification #96 (5-kg load) since ceramic fracture along the margin of the implant abutments was reported in pilot studies when loaded with weight greater than 2 kg [14]. The excess cement was removed by a scaler after the initial setting. For the cementation with TempBond (control), after drying the abutment and the internal surface of the restorations, equal lengths of base and accelerator were extruded onto the mixing pad. The paste was thoroughly mixed for approximately 30 seconds and a thin layer of the mixed cement was applied to the internal surfaces of the restoration. The restoration was firmly seated on the abutment. After the material set, the excess material was removed with a scaler and the color was re-measured after final setting of the luting agent. The color measurements were all performed by a single skilled operator blinded to the subgroups. After assessing color and measuring dislodgment force for each crown, the abutment was cleaned and the next crown was cemented.

## Aging process

The specimens were stored at 37°C in 100% humidity for 24 hours and then subjected to 5000 thermal cycles [2] between 5° and 55° with a dwell time of 10 seconds (TC-300; Vafaei

**Table 1. Type, manufacturer, composition (%) and LOT number of the cements.**

| Cement | Cement type | Manufacturer | Compositions | LOT number |
|---|---|---|---|---|
| TempBond | Zinc oxide eugenol | Kerr | TEMP-BOND NE BASE: zinc oxide (60–100%), White mineral oil (petroleum) (5–10%), | 6987471 |
| | | | Temp-Bond Accelerator: Eugenol (30–60%) | |
| Temp-Bond Clear | Dual curable temporary resin cement | Kerr | Dibutyl phthalate (5–10%), | 7326030 |
| | | | Hydroxyethyl methacrylate (5–10%), Fumed silica (1–5%), | |
| | | | N-(2-Pyridyl) thiourea (1–5%), | |
| | | | Ethyldimethylaminobenzoate | |
| | | | (0.5–1.5%), Triclosan (0.5–1.5%). | |
| Implantlink Semi | Resin-based temporary luting cement | DETAX | aliphatic urethane acrylate(10 - < 15%), 1-benzyl-5-phenyl-hexahydropyrimidine-2,4,-6-trione (5 - < 10%), 1,6-hexanediol dimethacrylate(5 - < 10%), 2-hydroxyethyl methacrylate(< 1%), triclosan; 2,4,4'-trichloro-2'-hydroxy-diphenyl-ether; 5-chloro-2-(2,4-dichlorophenoxy)phenol (< 1%), 2-Ethylhexyl 4-(dimethylamino)benzoate(< 1%). | 220804 |

Industrial, Tehran, Iran). The color was, then, re-measured to check the effect of aging on the cement color.

The color difference was calculated by using the CIEDE2000 formula [15]:

$$\Delta E_{00}^{*} = \sqrt{\left(\frac{\Delta L'}{k_L S_L}\right)^2 + \left(\frac{\Delta C'}{k_C S_C}\right)^2 + \left(\frac{\Delta H'}{k_H S_H}\right)^2 + R_T \frac{\Delta C'}{k_C S_C}\frac{\Delta H'}{k_H S_H}}$$

The pre- and post-cementation color difference was called $\Delta E_{00}(1)$ and the post-cementation and post-aging color difference was defined as $\Delta E_{00}(2)$.

The perceptibility threshold was set at $\Delta E_{00} \leq 1.30$ and the clinical acceptability threshold was set at $\Delta E_{00} > 2.25$ [15].

## Pull-out test

The specimens were attached to the universal testing machine (Zwick/Roell Z020, Ulm, Germany) and subjected to a tensile force at a crosshead speed of 5 mm/min [16]. The retentive force causing the detachment of specimens was recorded in Newton. The pull-out tests were done by the same blinded operator. The specimens were examined by using a light microscope and their failure mode was classified as an adhesive (complete separation of the cement from the abutment or the crown), cohesive (failure within the cement), and mixed (a combination of adhesive and cohesive). Fracture of the crowns or abutments was also assessed.

## Statistical analysis

The data were analyzed by using SPSS software (IBM SPSS Statistics for Windows, v22.0; IBM Corp., IL, USA). Normal distribution was tested with the Kolmogorov-Smirnov test ($P<0.05$ indicates lack of normality). Kruskal-Wallis, Dunn's post-hoc, and Mann-Whitney non-parametric tests were used to compare $\Delta E_{00}(1)$ and $\Delta E_{00}(2)$ among the study groups and sub-groups, because the data for these two variables ($\Delta E00(1)$ and $\Delta E00(2)$) were not normal. After the normality test showed that the data for retentive strength was normal, the mean retentive strength was analyzed with two-way ANOVA followed by one-way ANOVA and Tukey's HSD post hoc test to determine statistically significant differences among the three employed cements. T-test was used to compare the retention of ceramic crowns as a function of the type of cement. To consider the effect of inflated type 1 error through Bonferroni correction, the P value obtained by Mann-Whitney U or T test was multiplied by 3 (number of comparisons). $P<0.05$ was considered to be statistically significant in all tests.

## Results

### Color change ($\Delta E_{00}$)

Kruskal-Wallis test revealed that unlike $\Delta E_{00}(2)$, $\Delta E_{00}(1)$ was significantly different among the types of cement with both monolithic zirconia ($P = 0.004$) and lithium disilicate ceramic crowns ($P = 0.002$) (Table 2, Fig 3). Pairwise comparison of the cements through Dunn's post-hoc test revealed that in monolithic zirconia crowns, the mean $\Delta E_{00}(1)$ in TempBond was significantly higher than that in Temp-Bond Clear ($P = 0.007$) and Implantlink Semi cement ($P = 0.020$). Similarly, in lithium disilicate ceramic crowns, TempBond had a significantly higher mean $\Delta E_{00}(1)$ than Implantlink Semi cement ($P<0.001$) (Table 2).

Mann-Whitney test compared the mean $\Delta E_{00}(1)$ and $\Delta E_{00}(2)$ among the cement groups as a function of the type of ceramic crown. Accordingly, only in presence of Implantlink Semi cement, $\Delta E_{00}(1)$ of monolithic zirconia was significantly higher than that of the lithium

**Table 2. Mean±SD and median [minimum, maximum] of $\Delta E_{00}(1)$ and $\Delta E_{00}(2)$ and the results of Kruskal-Wallis H test and Dunn's post-hoc test comparisons of the cements.**

| Type of restoration | $\Delta E_{00}$ | Type of cement | | | P value |
|---|---|---|---|---|---|
| | | TempBond | Temp-Bond Clear | Implantlink Semi | |
| Monolithic Zirconia | $\Delta E_{00}(1)$ | 3.57±0.19 | 2.12±0.35 | 2.41±0.15 | 0.004[¶] |
| | | 3.50 [3.10,4.60] | 2.10[0.8,3.60] | 2.20[1.90,2.90] | |
| | P Value | 0.007[ac©] | 0.999[ab©] | 0.020[bc©] | |
| | $\Delta E_{00}(2)$ | 0.83±0.18 | 0.93±0.35 | 1.13±0.21 | 0.509[¶] |
| | | 0.80[0.20,1.40] | 0.50[0.2,2.5] | 1.10[0.30,2.0] | |
| | P value | — | —— | — | |
| Lithium disilicate | $\Delta E_{00}(1)$ | 4.91±0.66 | 2.08±0.54 | 0.78±0.18 | 0.002[¶] |
| | | 4.3[2.90,7.40] | 2.30[0.30,4.10] | 0.70[0.20,1.70] | |
| | P value | 0.067[ac©] | 0.635[ab©] | 0.001[bc©] | |
| | $\Delta E_{00}(2)$ | 1.11±0.35 | 1.46±0.46 | 1.40±0.31 | 0.735[¶] |
| | | 0.70[0.2,2.50] | 1.0[0.30,3.30] | 1.5[0.30,2.40] | |
| | P value | ——— | ———— | ———— | |

ab: Temp-Bond Clear vs. Implantlink Semi

ac: Temp-Bond Clear vs. TempBond

bc: Implantlink Semi vs. TempBond

¶: Results Of Kruskal-Wallis H test

©: Results of Dunn's post-hoc test comparisons of the cements

disilicate ceramic ($P = 0.003$). Yet, $\Delta E_{00}(2)$ was not significantly different between the two ceramics with any of the employed cements (Table 3).

## Retentive strength

Two-way ANOVA showed that the retentive strength was significantly influenced by both the type of ceramic crown ($P = 0.018$) and cement ($P = 0.016$). Interaction between the ceramic

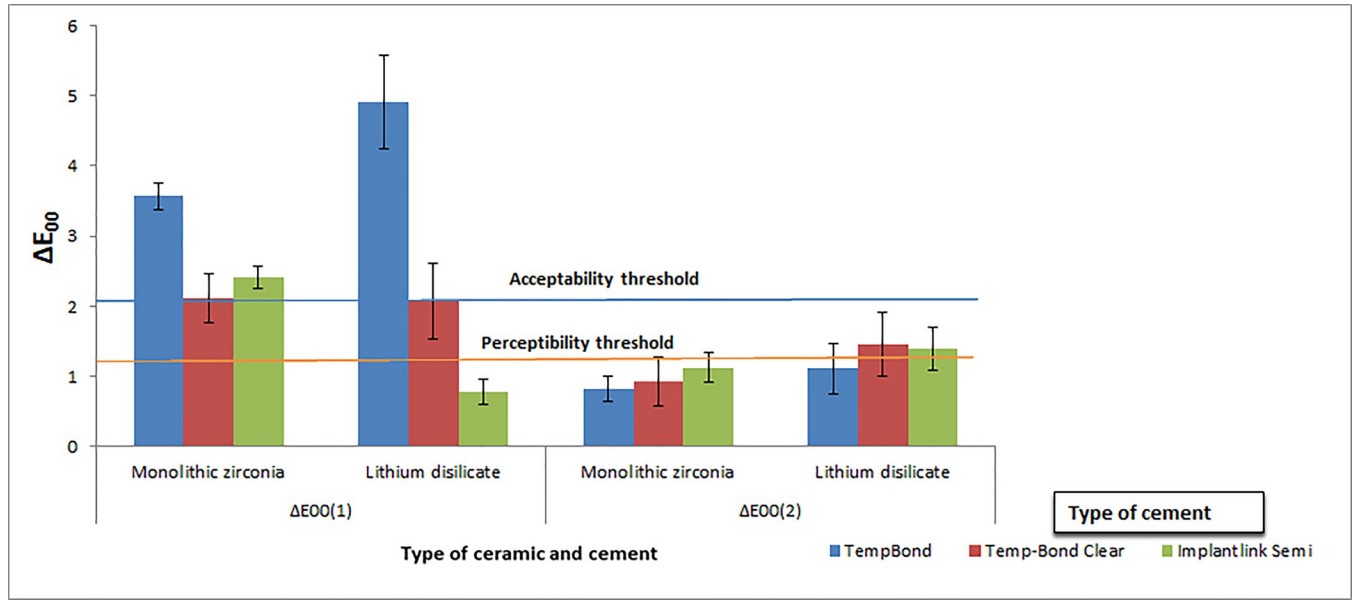

**Fig 3. Mean and standard deviation of $\Delta E_{00}(1)$ and $\Delta E_{00}(2)$.**

**Table 3. Results of Mann-Whitney test for $\Delta E_{00}$ differences between ceramic restorations in each temporary cement.**

| Cement | $\Delta E_{00}$ | Ceramic | P value |
|---|---|---|---|
| TempBond | $\Delta E_{00}(1)$ | Monolithic zirconia | 0.777 |
| | | Lithium disilicate | |
| | $\Delta E_{00}(2)$ | Monolithic zirconia | >0.999 |
| | | Lithium disilicate | |
| Temp-Bond Clear | $\Delta E_{00}(1)$ | Monolithic zirconia | >0.999 |
| | | Lithium disilicate | |
| | $\Delta E_{00}(2)$ | Monolithic zirconia | 0.627 |
| | | Lithium disilicate | |
| Implantlink Semi | $\Delta E_{00}(1)$ | Monolithic zirconia | 0.003 |
| | | Lithium disilicate | |
| | $\Delta E_{00}(2)$ | Monolithic zirconia | >0.999 |
| | | Lithium disilicate | |

crown and cement type was also significant ($P < 0.001$) (Table 4). Meanwhile, one-way ANOVA showed significant differences among the three cements in both monolithic zirconia ($P = 0.001$) and lithium disilicate ($P = 0.002$) ceramic restorations (Table 5).

Tukey's HSD post hoc test showed that in the monolithic zirconia group, TempBond created significantly more retention than TempBond Clear ($P = 0.003$) and Implantlink Semi cement ($P = 0.001$). However, in the lithium disilicate ceramic group, the specimens cemented with Implantlink Semi cement were significantly more retentive than those cemented with TempBond Clear ($P = 0.019$) and TempBond ($P = 0.001$) (Table 5, Fig 4).

Table 7 displays the pattern of cement distribution after the removal of cemented crowns. None of the crowns broke during applying dislodgment force; nor was any of the zirconia superstructures of the customized zirconia abutments separated from the titanium inserts.

## Discussion

The present findings rejected the null hypothesis since the type of provisional resin cement and ceramic restoration affected the final color of ceramic crowns and retentive strength of implant-supported all-ceramic restorations.

### Luting cement and color ($\Delta E_{00}(1)$)

This study showed that the provisional resin cement significantly improved the color match of monolithic zirconia and lithium disilicate ceramic restorations over the zirconia abutment by reducing the $\Delta E_{00}(1)$ values compared with the control group. Considering the previously established clinical perceptibility and acceptability thresholds [15], the color change for the control group was clinically unacceptable ($\Delta E_{00} > 2.25$) for both ceramic restorations, and also unacceptable for Implantlink Semi cement in the monolithic zirconia group ($\Delta E_{00} > 2.25$).

**Table 4. Two-way ANOVA results for retentive strength (N).**

| | Type III Sum of Squares | Df | Mean square | F | Sig. |
|---|---|---|---|---|---|
| Type of ceramic | 866.050 | 1 | 866.050 | 6.128 | .018 |
| Type of cements | 1311.627 | 2 | 655.813 | 4.641 | .016 |
| Ceramic×cements | 5888.943 | 2 | 2944.471 | 20.836 | < .001 |

**Table 5. Mean and standard deviation and pairwise comparison of retentive strength (N).**

| Ceramic | Cement | Mean | SD | P* | Tukey's HSD post hoc test | P¶ |
|---|---|---|---|---|---|---|
| Monolithic zirconia | TempBond | 28.20 | 6.75 | 0.001 | TempBond vs. Temp-Bond Clear | 0.003 |
| | Temp-Bond Clear | 17.40 | 3.66 | | Temp-Bond Clear vs. Implantlink Semi | 0.659 |
| | Implantlink Semi | 11.69 | 8.00 | | Implantlink Semi vs. TempBond | 0.001 |
| Lithium disilicate | TempBond | 8.03 | 4.67 | 0.002 | TempBond vs. Temp-Bond Clear | 0.151 |
| | Temp-Bond Clear | 24.30 | 12.77 | | Temp-Bond Clear vs. Implantlink Semi | 0.019 |
| | Implantlink Semi | 49.51 | 23.24 | | Implantlink Semi vs. TempBond | 0.001 |

P*: results of one-way ANOVA

P¶: results of Tukey's HSD post hoc test

The color of ceramic restorations is reported to be affected by the luting cement [12, 17, 18]; as Carrabba et al. [17] documented imperceptible color change only with the clear shade of resin cements in combination with the thickest CAD/CAM feldspathic ceramic (2 mm). Tabatabaian et al. [18] detected that zinc phosphate and TempBond cements caused unacceptable color changes on a zirconia framework; while, a glass ionomer and resin cement led to acceptable results. However, Malkondu et al. [12] found the lowest and highest ΔE in monolithic zirconia crowns cemented with resin-modified glass ionomer and zirconia crowns cemented with resin cement, respectively.

Although the provisional resin cement used in this study significantly improved the color match of both types of ceramic restorations, it was still clinically unacceptable for Implantlink Semi cement in monolithic zirconia group ($\Delta E_{00} > 2.25$); but, using this cement with lithium disilicate resulted in a great color match ($\Delta E_{00} < 1.3$). However, it should be noted that the final color of all-ceramic restorations might be unpredictable after the use of temporary cements, as

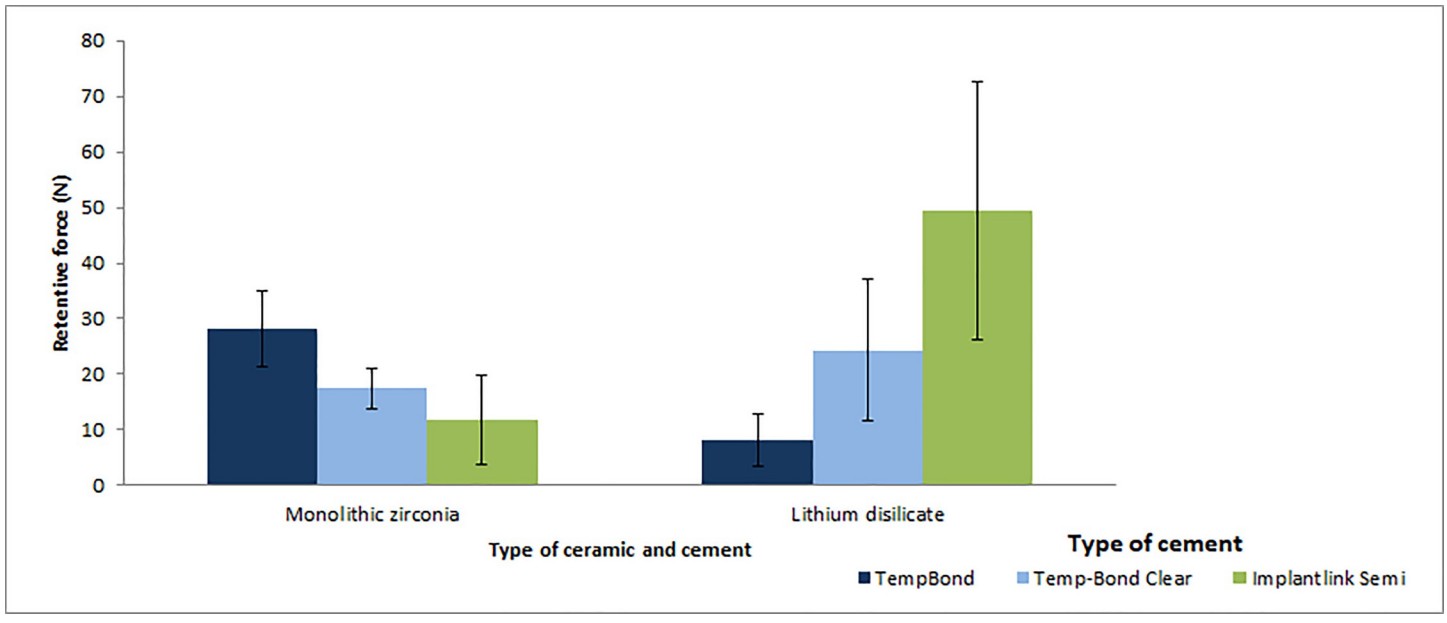

**Fig 4. The mean and standard deviation of retentive strength of cements as a function of the type of restorations.** Concerning the effect of ceramic restoration, the mean retention was significantly different between the two ceramic types only when cemented with TempBond ($P < 0.001$) and Implantlink Semi cement ($P = 0.004$); that is, the mean retentive strength of TempBond was significantly higher in monolithic zirconia restorations; while in lithium disilicate, Implantlink Semi was more retentive (Table 6).

**Table 6. Comparing the mean retentive strength of ceramic crowns as a function of the type of cement.**

| Type of cement | Type of restoration | Mean | SD | P value |
|---|---|---|---|---|
| TempBond | Monolithic zirconia | 28.20 | 6.76 | < .001 |
| | Lithium disilicate | 8.03 | 4.67 | |
| Temp-Bond Clear | Monolithic zirconia | 14.70 | 3.67 | 0.098 |
| | Lithium disilicate | 24.30 | 12.77 | |
| Implantlink Semi | Monolithic zirconia | 11.69 | 8.01 | 0.004 |
| | Lithium disilicate | 49.51 | 23.24 | |

reported by Liu et al. [19]. They also suggested that for a 1-mm-thick high or ultrahigh-translucent all-ceramic restoration, non-opaque or natural opaque shade cements can improve the color match before and after cementation more than the opaque cements did; however, it might be insufficient to make the final color clinically acceptable. The role of cement is even more imperative in more translucent ceramics restorations, where opaque cements create a lighter shade and transparent cements create a darker shade due to changes in $L^*$ values [20].

The effect of luting cement on the color of an all-ceramic restoration is in association with the thickness of the ceramic and cement, as well as the cement shade [13, 21]. Tabatabaian et al. [13] defined the minimum thickness of 1 mm for acceptable masking ability and a minimum thickness of 1.6 mm for the ideal masking ability of a zirconia ceramic on a black and white substrate. In addition, Vichi et al. [21] showed that differences in cement thickness (0.1 or 0.2 mm) slightly affected the final color of an all-ceramic crown. Since the current study used ceramic and cement in a single thickness (1.0 and 0.04 mm, respectively), no conclusion can be drawn regarding the ceramic and cement thicknesses.

The wide variety of color measurement methods restricts the comparison between studies. Tsiliagkou et al. [22] assessed the repeatability and accuracy of Easyshade (Vita), SpectroShade (MHT Optic Research), and ShadeVision (XRite) dental spectrophotometers and reported SpectroShade as the most accurate and reliable of the three color-matching devices. It was corroborated by Mehl et al.'s [23] and Khurana et al.'s [24] studies. However, Dozic et al. reported Easyshade (the one used in the present study) as the most reliable instrument of shade matching both in vitro and in vivo [25]. The difference among the studies can also be due to the employed color difference formulae. Although some researchers might still use the CIE 76 formula, CIEDE2000 is suggested as a more applicable and reliable formula in dentistry [26].

## Aging and the final color ($\Delta E_{00}(2)$)

The present study also assessed the effect of aging and thermocycling on the final color of all-ceramic restorations. Concerning the aging, although the post-aging $\Delta E_{00}(2)$ did not increase as much as $\Delta E_{00}(1)$, it was above the clinically perceptible threshold in lithium disilicate crowns cemented with Implantlink Semi and Temp-Bond Clear cements; indicating the mild

**Table 7. Frequency of mode of failure in different groups (%).**

| Type of cement/crown | Adhesive (cement-abutment interface) | Adhesive (cement-crown interface) | Cohesive (within the cement) | Mixed |
|---|---|---|---|---|
| TempBond / monolithic zirconia | 0 | 0 | 0 | 100 |
| Temp-Bond Clear / monolithic zirconia | 0 | 100 | 0 | 0 |
| Implantlink Semi / monolithic zirconia | 20 | 80 | 0 | 0 |
| TempBond / lithium disilicate | 100 | 0 | 0 | 0 |
| Temp-Bond Clear / lithium disilicate | 50 | 30 | 0 | 20 |
| Implantlink Semi / lithium disilicate | 70 | 0 | 0 | 30 |

influence of cement type on pre- and post-aging $\Delta E_{00}$, to be considered by the clinicians. Gürdal et al. [27] reported that the aging process increased the $b^*$ value. Gradual changes of camphorquinone towards yellow have also been reported [28, 29], as it is possible in the newly developed photoinitiator in the tested resin cements, which is yellow in color [27]. In the current study, increasing the number of cycles (5000 times between 5˚ and 55˚ with a dwell time of 10 seconds was applied) might have had resulted in more intense color changes. Mesbah et al. [30] noted an increase in $\Delta E$ in all cements by increasing the number of thermal cycles.

## Retentive strength (N)

The present study also found that the type of cement and ceramic significantly influenced the retention. The mean retention strength decreased in the monolithic zirconia crowns as the cement changed from TempBond to Temp-Bond Clear and then to Implantlink Semi cement; whereas, quite the opposite trend occurred in the lithium disilicate ceramic crowns.

Sarfaraz et al. [8] cemented the metal copings on metal abutments by using three cements and observed that the retentive strength was the highest in non-eugenol temporary resin cement, followed by non-eugenol zinc oxide cement, and resin-based acrylic urethane cement, respectively. They concluded that non-eugenol temporary resin cement might be better for cementation of implant prosthesis due to its superior mechanical properties.

Manufacturers of some new provisional resin cements claim to have higher retentive strength [6]. It has been corroborated by some previous [7, 8] and the present study in the lithium disilicate ceramic group; although, some other research documented the opposite, even compared with other temporary cements [31]. However, resin-based provisional cement has certain advantages such as ease of retrievability with adequate strength and excess cement removal, and excellent marginal adaptability [32, 33].

As the manufacturers recommended avoiding the surface preparation on the abutments or intaglio surface of the crowns, the two currently-employed resin-based provisional cements used mechanical retention to adhere the crown to the abutment. Although the inner surface of both crown groups was sandblasted, they were not prepared on the abutments. Recent reports have shown that the bond strength of zirconium oxide ceramics could be improved only by airborne particle abrasion on the ceramic surface and the use of a composite resin cement containing an adhesive phosphate monomer [34, 35]. Nejatidanesh et al. [16] reported that silicoating improved the retentive strength of zirconia copings more than aluminum oxide airborne particle abrasion. The alloy primers cannot improve the bond of temporary resin cements since these cements lack 10-methacryloyloxydecyl dihydrogen phosphate, which exists in many resin cements like Panavia F [36].

Among the limitations of this study was the 0.04-mm cement space, which might have compromised the retentive properties of the resin-based luting cements, as a higher film thickness would have compromised their physical properties. Moreover, using a constant removing force might have affected the results, since intraoral occlusal forces have a dynamic nature and not a monotonic static load; therefore, cement behavior might be different under fatigue loading rather than a static force load. Nor did the current study measure the retentive strength before aging. Further studies are recommended to assess the effect of aging and methods of surface treatment on the retentive strength of currently-studied resin-based temporary cements.

## Conclusions

Within the limitations of this in-vitro study, it can be concluded that:

1. In the monolithic zirconia crowns, the mean $\Delta E_{00}(1)$ of TempBond (control) was higher than TempBond Clear and Implantlink Semi. Similarly, in the lithium disilicate ceramic group, $\Delta E_{00}(1)$ of TempBond was higher than Implantlink Semi.

2. The mean retentive strength in the monolithic zirconia group was the highest for Temp-Bond (control), followed by TempBond Clear, and Implantlink Semi cement. However, for the lithium disilicate ceramic crowns, the mean retentive strength was the highest for Implantlink Semi cement, followed by TempBond Clear, and TempBond.

## Supporting information

**S1 Table. Raw data of $\Delta E_{00}(1)$ and $\Delta E_{00}(2)$.**
(XLSX)

**S2 Table. Raw data of retentive strength.**
(XLSX)

## Acknowledgments

This article was based on the postgraduate thesis by Dr. Seyede Mina Salehi Dehno. The authors would like to thank Dr. Vossoughi from the Dental Research Development Center of School of Dentistry for the statistical analyses and Miss Farzaneh Rasooli for proofreading, copyediting, and improving the use of English in this manuscript. Appreciations are also expressed to ArmanSalamat Dental laboratory and Mr. Abdi for the preparation of the laboratory specimens.

## Author Contributions

**Conceptualization:** Seyede Mina Salehi Dehno, Rashin Giti, Mohammad Hassan Kalantari, Farhad Mohammadi.

**Data curation:** Seyede Mina Salehi Dehno, Rashin Giti, Mohammad Hassan Kalantari, Farhad Mohammadi.

**Formal analysis:** Mohammad Hassan Kalantari, Farhad Mohammadi.

**Funding acquisition:** Rashin Giti.

**Investigation:** Seyede Mina Salehi Dehno, Mohammad Hassan Kalantari.

**Methodology:** Seyede Mina Salehi Dehno, Rashin Giti, Mohammad Hassan Kalantari, Farhad Mohammadi.

**Project administration:** Rashin Giti.

**Resources:** Rashin Giti, Mohammad Hassan Kalantari.

**Software:** Seyede Mina Salehi Dehno, Mohammad Hassan Kalantari, Farhad Mohammadi.

**Supervision:** Mohammad Hassan Kalantari.

**Validation:** Seyede Mina Salehi Dehno, Rashin Giti, Mohammad Hassan Kalantari, Farhad Mohammadi.

**Visualization:** Seyede Mina Salehi Dehno, Rashin Giti, Mohammad Hassan Kalantari, Farhad Mohammadi.

**Writing – original draft:** Seyede Mina Salehi Dehno.

**Writing – review & editing:** Seyede Mina Salehi Dehno, Rashin Giti, Mohammad Hassan Kalantari, Farhad Mohammadi.

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
