## [Decision Letter · Decision Letter 0]

2 Nov 2021

PONE-D-21-30628The effects of provisional resin cements on the color and retentive strength of all-ceramic restorations cemented on customized zirconia abutmentsPLOS ONE

Dear Dr. Giti,

Thank you for submitting your manuscript to PLOS ONE. After careful consideration, we feel that it has merit but does not fully meet PLOS ONE’s publication criteria as it currently stands. Therefore, we invite you to submit a revised version of the manuscript that addresses the points raised during the review process. Please, address all the comments made by the reviewers.

We look forward to receiving your revised manuscript.

Kind regards,

Antonio Riveiro Rodríguez, PhD

Academic Editor

PLOS ONE

Journal Requirements:

Reviewers' comments:

Reviewer's Responses to Questions

**Comments to the Author**

1. Is the manuscript technically sound, and do the data support the conclusions?

Reviewer #1: Yes

Reviewer #2: Yes

2. Has the statistical analysis been performed appropriately and rigorously? 

Reviewer #1: No

Reviewer #2: I Don't Know

3. Have the authors made all data underlying the findings in their manuscript fully available?

Reviewer #1: No

Reviewer #2: Yes

4. Is the manuscript presented in an intelligible fashion and written in standard English?

Reviewer #1: Yes

Reviewer #2: Yes

5. Review Comments to the Author

Reviewer #1: This in vitro research assesses the effect of two different provisional resin cements (ie, TempBond Clear and Implantlink Semi) on the color and retentive strength of implant crowns manufactured from two different ceramics (ie, monolithic zirconia and lithium disilicate). Crowns cemented with conventional zinc oxide-eugenol cement (ie, TempBond) served as control.

The study is of interest to the scientific community. However, statistical analysis and reporting should comply with standards regarding the publication of results derived from material laboratory research. Therefore, the following issues should be corrected:

- TempBond is a conventional zinc oxide-eugenol cement and NOT a provisional resin cement. Only two different provisional resin cements (ie, TempBond Clear and Implantlink Semi) were used while TempBond served as control. Please make this clear throughout the whole manuscript.

- Which delivery option of TempBond was used (ie, automix syringe, tube delivery, or unidose)?

- What was your rationale for the low number of thermocycles (ie, 500)? The applied number of cycles corresponds to about 2.6 weeks clinical service (Gale and Darvell 1999).

- How did you define the required number of specimens? Was any sample size calculation performed?

- Were fabricated crowns randomly assigned to each treatment group?

- How many operators performed the experiments?

- Was the operator of the testing machine blinded regarding the treatment group?

- Please supply LOT numbers of all materials used and provide a table depicting each cements' composition (%).

- Please use uniform spelling/capitalization of all materials throughout the manuscript's main text and figures.

- Please provide the Weibull distribution parameters (Weibull modulus and characteristic strength) of the retentive strength.

- Were all data distributed normally? You only state that normal distribution was tested with Kolmogorov-Smirnov test. However, subsequently both non-parametric tests and parametric tests were used.

- Results from Mann–Whitney U test were not corrected for multiple testing.

Reviewer #2: This research is under the scope of this journal; the topic is relevant for readers, and this research deals with potentially significant knowledge to the field.

However, there are some concerns in the about the present manuscript:

Introduction

(Statement of Relevance) What is the importance of this study for the clinical? You do not think this study is included in the others already done? Which results are comparable? What this study has new?

Materials and Methods

This section would better communicate to readers if restructured. A flowchart or diagram of the experimental processing would be valuable.

How was the sample calculated? Did the authors perform a power analysis to evaluate if this sample size was appropriate?

When mentioning materials or devices: for some of them you don't mention the manufacturer at all, for some you mention only the manufacturer, for some the manufacturer and city, for some you mention the manufacturer and city/ country. Standarized these presentation.

Line 129 - What was the manufacturer's instruction at the time?

Please confirme the tensile force at a crosshead speed of 5 mm/min or 0.5 mm/min. Please read this article(https://doi.org/10.1007/s00784-020-03640-7) or https://doi.org/10.1590/S1678-77572009000600012.

Results

- Improve the resolution quality of all figures and graphs (and a presentation). The font/ language in the figure/caption is different from the text. Please, standardized the size and the font in the figures and charts with the font of the manuscript. 

Conclusions

The conclusion section should more thoroughly summarize the results, this section is too long and sometime too ambiguous.

6. PLOS authors have the option to publish the peer review history of their article (what does this mean?). If published, this will include your full peer review and any attached files.

Reviewer #1: No

Reviewer #2: No

---

## [Author Response · Author response to Decision Letter 0]

12 Nov 2021

Reviewer #1

1) TempBond is a conventional zinc oxide-eugenol cement and NOT a provisional resin cement. Only two different provisional resin cements (ie, TempBond Clear and Implantlink Semi) were used while TempBond served as control. Please make this clear throughout the whole manuscript.

Response: According to the reviewer's precise comment, the control group was clearly defined throughout the manuscript.

2) Which delivery option of TempBond was used (ie, automix syringe, tube delivery, or unidose)?

Response: The tube delivery of TempBond was used and completely described in the revised manuscript.

3) - What was your rationale for the low number of thermocycles (ie, 500)? The applied number of cycles corresponds to about 2.6 weeks clinical service (Gale and Darvell 1999).

Response: The number of thermal cycles was mistyped as 500 instead of 5000 cycles. It was corrected in the revised manuscript and respective citation was added.

4) - How did you define the required number of specimens? Was any sample size calculation performed?

Response: Although the sample size had not been statistically calculated, the tests resulted in significant p-values where the effect sizes were moderate to large, indicating that the sample size in each category was sufficient to interpret the results.

5) - Were fabricated crowns randomly assigned to each treatment group?

Response: yes, the fabricated crowns were randomly assigned to each cement group. 

6) - How many operators performed the experiments?

Response: As mentioned in the revised manuscript, all color measurements and pull-out tests were performed by a single skilled operator.

7) - Was the operator of the testing machine blinded regarding the treatment group?

Response: Yes, the operator of the testing machine was blinded to the treatment groups.

8) - Please supply LOT numbers of all materials used and provide a table depicting each cements' composition (%).

Response: According to the reviewer's precise comment, a new table was added to mention the LOT number, type and manufacturer of all cements. However, since the composition of provisional resin cements (%) was not clearly reported by the manufacturers, such data could not be included in the table.

9) - Please use uniform spelling/capitalization of all materials throughout the manuscript's main text and figures.

Response: The spelling/capitalization was uniformly revised all over the text and figures.

10) - Please provide the Weibull distribution parameters (Weibull modulus and characteristic strength) of the retentive strength.

Response: The Weibull distribution parameters were not reported in the present study as in the preceding research.

11) - Were all data distributed normally? You only state that normal distribution was tested with Kolmogorov-Smirnov test. However, subsequently both non-parametric tests and parametric tests were used.

Response: In the statistical analysis section, it was stated that Kolmogorov-Smirnov test was used to test normality assumption. The two variables of ΔE00(1) and ΔE00(2) were not normally distributed; hence, non-parametric tests were used for the comparisons. Distribution of retentive strength was normal; thus, parametric tests were employed for the statistical comparisons. The revised manuscript clearly explains the tests used for each variable in.

- Results from Mann–Whitney U test were not corrected for multiple testing.

Response: The results of post-hoc tests (Dunn’s test for nonparametric and Tukey’s test for One-way ANOVA) were displayed in Tables 2 and 5 for each type of restoration. Moreover, the results of Mann-Whitney and t test (p-values) were adjusted based on the number of comparisons. Because the two ceramic restorations were separately compared in each cement (3 cements), the p-values were multiplied by 3 and reported in the revised tables.

Reviewer #2

Introduction

1)(Statement of Relevance) What is the importance of this study for the clinical? You do not think this study is included in the others already done? Which results are comparable? What this study has new?

Response: As mentioned in the Introduction section of the revised manuscript, limited information is available about the effect of provisional resin cements on the final color and retentive strength of implant-supported ceramic restorations. Nor has any study evaluated the effect of TempBond Clear and Implantlink Semi provisional resin cements on the retentive strength and final color of implant-supported all-ceramic restorations before and after thermocycling. 

Materials and Methods

2)This section would better communicate to readers if restructured. A flowchart or diagram of the experimental processing would be valuable.

Response: The structure of Materials and Methods section was revised and divided into subsections with definite subheadings.

3)How was the sample calculated? Did the authors perform a power analysis to evaluate if this sample size was appropriate?

Response: Although the sample size had not been statistically calculated, the tests resulted in significant p-values where the effect sizes were moderate to large, indicating that the sample size in each category was sufficient to interpret the results.

4)When mentioning materials or devices: for some of them you don't mention the manufacturer at all, for some you mention only the manufacturer, for some the manufacturer and city, for some you mention the manufacturer and city/ country. Standardized this presentation.

Response: With respect to the reviewer's meticulous suggestion, the manufacturer's detail for each material was standardized as trade name, manufacturer, country.

5)Line 129 - What was the manufacturer's instruction at the time?

Response: The manufacturer's instruction for all the three cements was described in the revised manuscript

6)Please confirm the tensile force at a crosshead speed of 5 mm/min or 0.5 mm/min. Please read this article(https://doi.org/10.1007/s00784-020-03640-7) or https://doi.org/10.1590/S1678-77572009000600012.

Response: The tensile force was applied at a crosshead speed of 5 mm/min as mentioned in the manuscript and the respective reference was cited.

Results

7) Improve the resolution quality of all figures and graphs (and a presentation). The font/ language in the figure/caption is different from the text. Please, standardized the size and the font in the figures and charts with the font of the manuscript. 

Response: The size and the font in the figures, charts, and body of the manuscript was standardized.

8)Conclusions

The conclusion section should more thoroughly summarize the results, this section is too long and sometime too ambiguous.

Response: The Conclusion section was thoroughly summarized in the revised manuscript.

---

## [Decision Letter · Decision Letter 1]

18 Nov 2021

PONE-D-21-30628R1The effects of provisional resin cements on the color and retentive strength of all-ceramic restorations cemented on customized zirconia abutmentsPLOS ONE

Dear Dr. Giti,

Thank you for submitting your manuscript to PLOS ONE. After careful consideration, we feel that it has merit but does not fully meet PLOS ONE’s publication criteria as it currently stands. Therefore, we invite you to submit a revised version of the manuscript that addresses the points raised during the review process.

Please, address all the comments made by the reviewers. As notice in the review reports, some of the issues given by reviewer 1 remain not satisfactorily answered. 

We look forward to receiving your revised manuscript.

Kind regards,

Antonio Riveiro Rodríguez, PhD

Academic Editor

PLOS ONE

Journal Requirements:

Reviewers' comments:

Reviewer's Responses to Questions

**Comments to the Author**

1. If the authors have adequately addressed your comments raised in a previous round of review and you feel that this manuscript is now acceptable for publication, you may indicate that here to bypass the “Comments to the Author” section, enter your conflict of interest statement in the “Confidential to Editor” section, and submit your "Accept" recommendation.

Reviewer #1: (No Response)

Reviewer #2: All comments have been addressed

2. Is the manuscript technically sound, and do the data support the conclusions?

Reviewer #1: Partly

Reviewer #2: Yes

3. Has the statistical analysis been performed appropriately and rigorously? 

Reviewer #1: No

Reviewer #2: Yes

4. Have the authors made all data underlying the findings in their manuscript fully available?

Reviewer #1: Yes

Reviewer #2: Yes

5. Is the manuscript presented in an intelligible fashion and written in standard English?

Reviewer #1: Yes

Reviewer #2: Yes

6. Review Comments to the Author

Reviewer #1: The authors have revised their manuscript. Some of the issues raised during the previous review round were answered / corrected. However, many issues still remain:

Former 3rd comment:

The number of thermocycles was not changed within the discussion section. The manuscript still reads 500 cycles ("500 times between 5° and 55° with a dwell time of 10 seconds was applied").

Former 4th comment:

Missing sample size calculations increase the risk of bias in laboratory research. Please see the scoring systems for the risk of bias assessment utilized within the following systematic reviews:

- Pires et al.: Is adhesive bond strength similar in primary and permanent teeth? A systematic review and meta-analysis. J Adhes Dent 2018; 20:87–97.

- Isolan et al.: Bonding to sound and caries-affected dentin: a systematic review and meta-analysis. J Adhes Dent 2018; 20:7–18.

- Soares et al.: Bovine tooth is a substitute for human tooth on bond strength studies: a systematic review and meta-analysis of in vitro studies. Dent Mater2016; 32:1385–93.

Former 5th comment:

For the above-mentioned reason, information regarding the random assignment of the fabricated crowns should be mentioned within the manuscript.

Former 7th comment:

For the above-mentioned reason, information regarding the blinded operator should be mentioned within the manuscript.

Former 8th comment:

Please extract composition (chemicals and %) from safety data sheets which are provided by the manufacturers.

Former 10th comment:

The missing Weibull parameters are a shortcoming of the present study. The authors' reply regardring this topic is unsatisfactory. Please see the following citations for further information regarding the importance of Weibull statistics within the context of dental materials research:

- McCabe and Carrick: A statistical approach to the mechanical testing of dental materials. Dent Mater 1986; 2:139-142.

- Quinn and Quinn: A practical and systematic review of Weibull statistics for reporting strengths of dental materials. Dent Mater 2010; 26:135-147.

Former 11th comment:

Information regarding the results of the Kolmogorov-Smirnov test are still missing within the text. Also, mentioned statistical test differ between the abstract and the main text. Also information regarding the meaning of the error bars shown in figures 3 and 4 is missing (eg, stand error, standard deviation).

I noticed the revised p-values derived from the Mann-Whitney U tests. Based on the authors' reply, a Bonferroni correction was utilized. However, this information is missing within the text.

Reviewer #2: This research is under the scope of this journal; the topic is interesting for readers and this research deals with potentially significant knowledge to the field and an open new way for future studies.

The authors improved the quality of the manuscript after the reviewer's indications.

7. PLOS authors have the option to publish the peer review history of their article (what does this mean?). If published, this will include your full peer review and any attached files.

Reviewer #1: No

Reviewer #2: No

---

## [Author Response · Author response to Decision Letter 1]

26 Nov 2021

Reviewer #1

1)The number of thermocycles was not changed within the discussion section. The manuscript still reads 500 cycles ("500 times between 5° and 55° with a dwell time of 10 seconds was applied").

Response: The number of thermal cycles was corrected in the Discussion section of revised manuscript.

2)Missing sample size calculations increase the risk of bias in laboratory research. Please see the scoring systems for the risk of bias assessment utilized within the following systematic reviews:

Response: Despite missing to calculate the sample size in advance, the statistical tests yienled significant p-values where the effect sizes were moderate to large. It indicated that the sample size in each category was sufficient in this study.

3) For the above-mentioned reason, information regarding the random assignment of the fabricated crowns should be mentioned within the manuscript.

Response: According to the reviewer's wise comment, it was stated in the revised manuscript that the fabricated crowns were randomly assigned to three cement subgroups.

4) For the above-mentioned reason, information regarding the blinded operator should be mentioned within the manuscript.

Response: The operator's blindness to the tested subgroups was clearly mentioned in the revised manuscript.

5) Please extract composition (chemicals and %) from safety data sheets which are provided by the manufacturers.

Response: All of the chemical compositions and % were extracted from safety data sheets and embedded in Table 1.

6)The missing Weibull parameters are a shortcoming of the present study. The authors' reply regarding this topic is unsatisfactory. Please see the following citations for further information regarding the importance of Weibull statistics within the context of dental materials research:

- McCabe and Carrick: A statistical approach to the mechanical testing of dental materials. Dent Mater 1986; 2:139-142.

- Quinn and Quinn: A practical and systematic review of Weibull statistics for reporting strengths of dental materials. Dent Mater 2010; 26:135-147.

Answer: With all due respect to the reviewer's wise comment, the present study missed performing the Weibull statistics.

7) Information regarding the results of the Kolmogorov-Smirnov test is still missing within the text. Also, mentioned statistical test differ between the abstract and the main text. Also, information regarding the meaning of the error bars shown in figures 3 and 4 is missing (eg, stand error, standard deviation). I noticed the revised p-values derived from the Mann-Whitney U tests. Based on the authors' reply, a Bonferroni correction was utilized. However, this information is missing within the text.

Answer: The information regarding the Kolmogorov-Smirnov test was completely added to the Statistical analysis section of revised manuscript, also the total statistical test was added to the Abstract section of revised manuscript. Besides, the legends of Figures 3 and 4 state that the means and standard deviations of each variable were shown in the figures.

As presented in the Statistical Analysis section, all the statistical teste used in the presented study are as follows: Normal distribution was tested with the Kolmogorov-Smirnov test. Kruskal-Wallis, Dunn’s post-hoc, and Mann-Whitney non-parametric tests were used to compare ΔE00(1) and ΔE00(2) among the study groups and subgroups, because the data for these two variables (ΔE00(1) and ΔE00(2)) were not normal. After the normality test showed that the data for retentive strength was normal, the mean retentive strength was analyzed with two-way ANOVA followed by one-way ANOVA and Tukey's HSD post hoc test to determine statistically significant differences among the three employed cements. T-test was used to compare the retention of ceramic crowns as a function of the type of cement. P<0.05 was considered to be statistically significant in all tests.

---

## [Decision Letter · Decision Letter 2]

19 Dec 2021

PONE-D-21-30628R2The effects of provisional resin cements on the color and retentive strength of all-ceramic restorations cemented on customized zirconia abutmentsPLOS ONE

Dear Dr. Giti,

Thank you for submitting your manuscript to PLOS ONE. After careful consideration, we feel that it has merit but does not fully meet PLOS ONE’s publication criteria as it currently stands. Therefore, we invite you to submit a revised version of the manuscript that addresses the points raised during the review process.

Notice that some minor revisions are still required in the manuscript. Please, address the remaining comments pointed out by the reviewer 1.

We look forward to receiving your revised manuscript.

Kind regards,

Antonio Riveiro Rodríguez, PhD

Academic Editor

PLOS ONE

Journal Requirements:

Reviewers' comments:

Reviewer's Responses to Questions

**Comments to the Author**

1. If the authors have adequately addressed your comments raised in a previous round of review and you feel that this manuscript is now acceptable for publication, you may indicate that here to bypass the “Comments to the Author” section, enter your conflict of interest statement in the “Confidential to Editor” section, and submit your "Accept" recommendation.

Reviewer #1: (No Response)

Reviewer #2: All comments have been addressed

2. Is the manuscript technically sound, and do the data support the conclusions?

Reviewer #1: Yes

Reviewer #2: Yes

3. Has the statistical analysis been performed appropriately and rigorously? 

Reviewer #1: No

Reviewer #2: Yes

4. Have the authors made all data underlying the findings in their manuscript fully available?

Reviewer #1: Yes

Reviewer #2: Yes

5. Is the manuscript presented in an intelligible fashion and written in standard English?

Reviewer #1: Yes

Reviewer #2: Yes

6. Review Comments to the Author

Reviewer #1: The presented manuscript significantly improved during the previous review rounds. Despite the missing Weibull analysis, the manuscript might be acceptable after addressing the following remaining issues:

1. Information regarding my previous 7th comment is still missing ("Based on the authors' reply, a Bonferroni correction was utilized. However, this information is missing within the text."). Please add information within the statistical analysis section, if and how p-values derived from Mann-Whitney U and t tests were adjusted for multiple testing (eg, Bonferroni correction, Bonferroni-Holm correction).

2. Table 2: The Lilliefors correction is related to the Kolmogorov-Smirnov test for normality. Therefore, the footnote "* Lilliefors Significance Correction is P<0.05" is misleading.

3. Table 2: "0.999a©b" should be written as "0.999ab©".

4. Tables 5: The footnote "Significance correction is P<0.05" not required.

5. Tables 3 and 6: Again, the footnote "Significance correction is P<0.05" not required if information regarding the level of significance and adjustment for multiple testing is given within the statistical analysis section.

6. Please write p-values as "<.001" instead of ".000".

Reviewer #2: (No Response)

7. PLOS authors have the option to publish the peer review history of their article (what does this mean?). If published, this will include your full peer review and any attached files.

Reviewer #1: No

Reviewer #2: No

---

## [Author Response · Author response to Decision Letter 2]

21 Dec 2021

Reviewer #1

1. Information regarding my previous 7th comment is still missing ("Based on the authors' reply, a Bonferroni correction was utilized. However, this information is missing within the text."). Please add information within the statistical analysis section, if and how p-values derived from Mann-Whitney U and t tests were adjusted for multiple testing (eg, Bonferroni correction, Bonferroni-Holm correction).

Response: Details of the employed Bonferroni correction was added to the statistical analysis section of revised manuscript and highlighted.

2. Table 2: The Lilliefors correction is related to the Kolmogorov-Smirnov test for normality. Therefore, the footnote "* Lilliefors Significance Correction is P<0.05" is misleading.

Response: This part was deleted as suggested by the reviewer. 

3. Table 2: "0.999a©b" should be written as "0.999ab©".

Response: It was corrected as recommended.

4. Tables 5: The footnote "Significance correction is P<0.05" not required.

Response: It was omitted according to reviewer's comment.

5. Tables 3 and 6: Again, the footnote "Significance correction is P<0.05" not required if information regarding the level of significance and adjustment for multiple testing is given within the statistical analysis section.

Response: It was omitted as suggested.

6. Please write p-values as "<.001" instead of ".000".he footnote "* Lilliefors Significance Correction is P<0.05" is misleading.

Response: The P-values were corrected according to the reviewer's meticulous comment.

---

## [Editor Report · Decision Letter 3]

30 Dec 2021

The effects of provisional resin cements on the color and retentive strength of all-ceramic restorations cemented on customized zirconia abutments

PONE-D-21-30628R3

Dear Dr. Giti,

We’re pleased to inform you that your manuscript has been judged scientifically suitable for publication and will be formally accepted for publication once it meets all outstanding technical requirements.

Kind regards,

Antonio Riveiro Rodríguez, PhD

Academic Editor

PLOS ONE

Additional Editor Comments (optional):

Authors have addressed all the comments made by the reviewer 1.

---

## [Editor Report · Acceptance letter]

7 Jan 2022

PONE-D-21-30628R3 

The effects of provisional resin cements on the color and retentive strength of all-ceramic restorations cemented on customized zirconia abutments 

Dear Dr. Giti:

I'm pleased to inform you that your manuscript has been deemed suitable for publication in PLOS ONE. Congratulations! Your manuscript is now with our production department. 

Kind regards, 

on behalf of

Dr. Antonio Riveiro Rodríguez 

Academic Editor

PLOS ONE